# Considerations in the Use of Gravitational Valves in the Management of Hydrocephalus. Some Lessons Learned with the Dual-Switch Valve

**DOI:** 10.3390/jcm10020246

**Published:** 2021-01-12

**Authors:** Maria A. Poca, Dario F. Gándara, Katiuska Rosas, Aloma Alcina, Diego López-Bermeo, Juan Sahuquillo

**Affiliations:** 1Department of Neurosurgery, Vall d’Hebron Hospital Universitari, Vall d’Hebron Barcelona Hospital Campus, Passeig Vall d’Hebron 119-129, 08035 Barcelona, Spain; dfgandara@vhebron.net (D.F.G.); krosas@vhebron.net (K.R.); aalcina@vhebron.net (A.A.); y2961706x@vhebron.net (D.L.-B.); sahuquillo@neurotrauma.net (J.S.); 2Neurotrauma and Neurosurgery Research Unit, Vall d’Hebron Institut de Recerca (VHIR), Vall d’Hebron Hospital Universitari, Vall d’Hebron Barcelona Hospital Campus, Passeig Vall d’Hebron 119-129, 08035 Barcelona, Spain; 3Department of Surgery, Universitat Autònoma de Barcelona, 08193 Bellaterra, Spain

**Keywords:** complications, Dual-Switch valve, gravitational valves, hydrocephalus, hydrostatic valves, shunt dysfunction

## Abstract

In the past decade, there has been a clear trend towards better outcomes in patients with hydrocephalus, especially those with normal pressure hydrocephalus (NPH). This is partly due to the availability of more sophisticated hardware and a better understanding of implants. However, there is little evidence to show the superiority of a specific type of valve over another. The most commonly reported consequence of hydrodynamic mismatch is shunt over-drainage. Simple differential pressure valves, with a fixed opening pressure or even adjustable valves, lead to non-physiologic intraventricular pressure (IVP) as soon as the patient moves into an upright posture. These valves fail to maintain IVP within physiological limits due to the changes in hydrostatic pressure in the drainage system. To solve this problem more complex third-generation hydrostatic valves have been designed. These gravitational devices aim to reduce flow through a shunt system when the patient is upright but there are important technical differences between them. Here we review the main characteristics of the Miethke^®^ Dual-Switch valve, which includes two valve chambers arranged in parallel: a low-opening pressure valve, designed for working in the supine position, and a second high-opening pressure valve, which starts working when the patient assumes the upright position. This paper specifies the main advantages and drawbacks of this device and provide a series of recommendations for its use. The discussion of this specific gravitational valve allows us to emphasize the importance of using gravitational control in implanted shunts and some the caveats neurosurgeons should take into consideration when using gravitational devices in patients with hydrocephalus. The correct function of any gravitational device depends on adequate device implantation along the vertical body axis. Misalignment from the vertical axis equal to or more than 45° might eliminate the beneficial effect of these devices.

## 1. Introduction

Hydrocephalus is a common disease treated in neurosurgical departments. Valve selection is an important factor for optimizing patient outcome. When differential-pressure valves (DPV) were introduced in the late 1950s, the neurosurgeon could select between high, medium, or low opening pressure valves. However, it soon became evident that over-drainage and its related complications (e.g., postural headaches, subdural bleeding, and subdural collections) were a direct consequence of the gravitationally-induced over-drainage when patients assume a standing or sitting position. The transition from the first generation of DPV to the more complex designs introduced later was a consequence of both neurosurgeons’ and engineers’ attempts to control the siphon effect that the DPV causes when the patient stands.

In the early 1970s, measurements of intracranial pressure (ICP) in patients revealed that pressure in the lateral ventricles was sub-atmospheric when the patient is erect [1]. The first DPV generation systematically induced a significant increase in the magnitude of negative ICP when patients were sitting or standing [1,2]. The negative shunt-induced ICP was attributed to the gravity-induced ‘siphoning’ effects, an intrinsic flaw of DPVs that were designed to work on the differential pressure between the ICP and the pressure where the distal catheter was placed, usually in the right atrium or in the peritoneum. Siphoning induces shunt over-drainage and is the cause of many reported adverse effects after shunting [3].

In recent decades, a wide variety of valves, shunt designs, and devices for siphon-control or gravity-compensating devices have been introduced [4]. Today, we must choose among valves with different opening pressures, variable hydrodynamics, and adjustable (‘programmable’) valves with or without incorporated ‘antisiphon’ or gravitational devices. The use of gravitational devices incorporated into the valve or added in series into the distal catheter can produce a more physiological cerebrospinal fluid (CSF) drainage through the shunt, reducing the complications of over-drainage. A pragmatic randomized controlled study in patients with idiopathic NPH (iNPH) showed that the use of a gravitational valve prevented one over-drainage complication in every third patient undergoing shunting compared with a non-gravitational programmable valve [5].

One of the first gravitational valves designed was the Miethke^®^ Dual-Switch valve (M-DSV; Aesculap, AG, Tuttlingen, Germany), which includes two valve chambers arranged in parallel: a low-opening pressure valve, designed for working in the supine position, and a second high-opening pressure valve, which starts working when the patient assumes the upright position. This device has been used for decades, especially in Europe and Japan [5]. The M-DSV has also been the main device used at our center since the second half of the 1990s. Subsequently, other smaller gravitational valves have displaced the use of the M-DSV in most centers.

The M-DSV is still available, is a good example of the gravitational device, and presents some differential characteristics that must be considered when selecting a valve for a specific patient. Here, we describe the characteristics of the M-DSV, its main advantages and drawbacks, and use our center learning curve with gravitational devices to make some recommendations for their use. The discussion of this gravitational valve allows us to emphasize the importance of using gravitational control in patients with hydrocephalus in which a shunt is implanted. We also want to discuss some of the issues that neurosurgeons should take into consideration when using gravitational devices in patients with hydrocephalus.

## 2. The Miethke^®^ Dual-Switch Valve

The M-DSV is a robust device with a titanium casing designed by biomedical engineers at the Technical University of Berlin in close cooperation with the neurosurgical department of the Virchow Klinikum, Free University Berlin (nowadays Charité, Humboldt University, Berlin, Germany) and the neurosurgical department of the University Hospital Essen. It is designed for subcutaneous implantation in the thoracic region of adult patients [6,7]. The device has a diameter of approximately 28 mm and is 6 mm thick, a thickness that is comparable to other valves and is distinctly thinner than a pacemaker, another device typically implanted subcutaneously in the thoracic region. Because of its design, subcutaneous pressure does not influence the M-DSV performance, as has been shown with other over-drainage-compensating devices, such as antisiphon valves [8]. M-DSV safety is also increased because it is better than other conventional valves for the drainage of viscous CSF, which might include proteins and cell debris [6,7]. This is because in the M-DSV, the area exposed to the fluid pressure is about 200-times larger than in other valves which systematically leads to higher acting (spring) forces which are consequently less affected by changed viscosity or debris [6]. The M-DSV includes two valve chambers arranged in parallel. The first, low-opening pressure valve is designed for working in the supine position, and the second, high-opening pressure valve starts working when the patient assumes the upright position. The pressure of both of these elements is controlled by ball-in-cone valves, and their main objective is to maintain ICP values within physiological ranges regardless of the patient’s posture (Figure 1) [7]. When patients are supine, the low-pressure chamber operates like a conventional DVP and is activated when the differential pressure is above the opening pressure (OP) of the valve. In the upright position, the low-pressure valve is closed by a tantalum ball. Afterward the control of the ventricular pressure in the upright position of the patient is shifted to the high-pressure-chamber of the M-DSV, and the postural changes move a tantalum ball [7] that opens the flow path for the high-pressure controller (Figure 1). The OP of this second valve is higher and takes into consideration the additional hydrostatic pressure of at least +35 cm H_2_O, which occurs in the upright position. The opening pressure for the upright position must be selected according to the height of the patient. At present, the M-DSV is available with an OP of 5–16 cm H_2_O for the supine position and 30–50 cmH_2_O for the upright position. For lumboperitoneal (LP) shunts, the available pressure settings for the lower-pressure valve is 10 cm H_2_O and for the upright position it is 30–50 cmH_2_O. Table 1 summarizes the pressures provided by the manufacturer.

## 3. Dual-Switch Valve Performance Reported by the Manufacturer

In in vitro tests carried out by the manufacturers, the M-DSV has shown good and predictable performance [6,7]. In terms of sensitivity to changes in CSF viscosity, composition, and sticking, as well as the influence of the pressure of the subcutaneous tissues, the M-DSV has superior performance compared with other valves [6,7]. Stepwise changes conducted in the laboratory–simulating changes from the upright to the supine position–demonstrated that ICP remains consistently within physiological limits [6]. The authors emphasize that all in vitro testing carried out provided strong evidence for the capability of the M-DSV to maintain intraventricular pressure within physiological ranges, regardless of changes in CSF flow and posture. The investigations substantiate the theoretical concept that the M-DSV is less affected by atypical CSF compositions in contrast to all other clinically used valves for treatment of hydrocephalus.

## 4. Independent Evaluation of the Performance of the Dual-Switch Valve

The UK Shunt Evaluation Laboratory assess the hydrodynamic properties of all types of shunts available in the UK [11,12]. This group has evaluated many valves available in the market, offering important and independent information that every neurosurgeon should consider when selecting a valve for a specific patient. The main aims of these authors have always been to: (1) assess hydrodynamic properties of a shunt regardless of the manufacturer; (2) check whether the shunt performs according to the manufacturer’s specifications and whether it complies with international standards (ISO/DIS7197); (3) characterize drainage capabilities with a more extensive range of tests than those listed in the ISO standard, and (4) summarize what is known from the literature about a shunt’s performance [9]. In these studies, the long-term stability of valve performance is tested in a laboratory environment that simulates the conditions within the human body, with the aim of demonstrating whether the shunt is susceptible to alterations in CSF drainage caused by postural changes, external pressure, change in ambient temperature, and/or the presence of a pulsating pattern in the inlet pressure, among other aspects [9].

In the report evaluation of the M-DSV, 11 valves were tested [9]. The authors demonstrated that M-DSV valves could limit drainage in a stepwise manner when the gravitational valve axis changed from horizontal to vertical. The performance was stable and durable, regardless of ambient pressure, temperature, or magnetic field. None of the parameters (OP, closing pressure (CP), and resistance of the shunt ()) were altered by changes in temperature (30–40 °C). Both OP and CP displayed minor variations (less than 1.5 mm Hg) in all tests. Good agreement of the operating pressures with the manufacturer’s data was recorded (differences of less than 1 mm Hg), and differences in measured parameters were minor among all tested valves. The valves did not show reflux when tested according to the ISO standard and did not exhibit a reversal of flow for an outlet-inlet differential pressure of up to 100 mm Hg [9,10]. Table 2 summarizes the technical findings reported by this group.

## 5. The Effect of MRI on the Function of M-DSV

The MR-compatibility of CSF shunts has become increasingly important with the growing number of high-field MR scanners employed and the new, powerful magnetic fields that will be available soon. Lindner et al. [13] evaluated three M-DSV valves before and after exposure to 3T MRI. The M-DSV does not have ferromagnetic properties because it is made of titanium, tantalum, and silicone [6,9]. In vivo, the valve is positioned in the thoracic region. All MRI experiments were performed on a 3T human-scale whole-body scanner (MAGNETOM Trio, Siemens, Erlangen, Germany) using a protocol analogous to clinical practice and with the valves mounted either on the side of a phantom aligned parallel to the axis of the main magnetic field or mounted perpendicularly [13]. Due to the chemical constituents of the valves, displacement or extensive heating of the devices was not expected, and the authors centered their evaluations on the functioning of the valves as the crucial parameter. The authors found that 3T magnetic fields did not influence the stability and safety of the M-DSV, with clear evidence that the valves continued to function correctly after MR scanning [13].

## 6. Pressure Selection of the Two Valves in the M-DSV

The M-DSV is available with several OPs for the supine position and for the upright position as listed in Table 1. The most reliable procedure for selecting a valve OP is based on the results of continuous ICP monitoring before surgery. In practice, however, when ICP recording is unavailable, most clinicians rely on the manufacturer’s recommendations [14]. According to the manufacturer, the recommended standard pressure setting for the lower-pressure valve (DualSwitch valve^®^, Instructions for use. C. Miethke GmbH & Co. KG, Potsdam, Germany) is 10 cm H2O (5 cm H_2_O for patients with NPH). The high-pressure side of the valve is calculated as a function of the patient’s height and is chosen so that with the patient upright, a ventricular pressure of at least −5 cm H_2_O is maintained (DualSwitch valve^®^, Instructions for use). Adequate pressure is calculated as follows: (1) measurement of the distance between the third ventricle (at the level of the foramen of Monro, roughly measured from the external auditory meatus) and the diaphragm (roughly measured at the level of the costal arch); (2) subtract 5 cm from the measured distance; (3) choose a valve with a high-pressure setting that exceeds the final measured value by the smallest amount (Figure 2). When these criteria are met, patient ICP would be expected to be maintained between −5 and +5 cmH_2_O even after the placement of the shunt [14].

Hertel et al. used a different strategy in which the opening pressure for the upright position (UPP) was dependent upon the patient’s height (180 cm or less, UPP = 30 cm or 40 cm H_2_O; height over 180 cm, UPP = 50 cm H_2_O) [15]. However, these authors did not consider the weight of the patient, which also determines the expected intra-abdominal pressure [16] and may explain the functional underdrainage found by some authors when using hydrostatic valves [17,18].

## 7. Intracranial Pressure Changes after M-DSV Implantation

The M-DSV was designed to adjust ICP according to the supine or upright position of the patient, avoiding over-drainage. However, switching between pressure settings is not gradual but instead occurs at an angle of approximately 60–70° [6]. Consequently, when changing patient position from horizontal to vertical, there is a short period in which ICP reaches a non-physiological negative value due to a delayed switching of the tantalum ball. Thus, if the patient changes slowly from the supine to the upright position, a situation of moderate over-drainage can occur [6]. In this period the tantalum ball only diminishes the flow throughout the valve as the ball starts to close the valve-seat. Although this phenomenon of temporary over-drainage has no serious clinical consequences, it explains the headaches that some patients might present during the first days after shunt implantation when they assume the sitting or standing position. To avoid headaches, during the first two days after valve implantation the patient’s bed should preferably be kept flat and not partly raised. Figure 3 illustrates the posture-induced ICP changes in a patient with idiopathic intracranial hypertension before and after the implantation of a 10/40 cm H_2_O M-DSV.

## 8. Surgical Aspects in the Implantation of the M-DSV

Antisiphon devices (ASDs) control over-drainage by increasing the resistance of the shunt (Rshunt). Gravitational devices (GDs) control over-drainage by increasing the OP of the valve through the movement of inbuilt metallic balls when the patient moves from the recumbent position. Consequently, the correct function of any GD depends on an adequate device’s vertical implantation. Any degree of deviation from the vertical axis equal to or more than 45° might eliminate the beneficial effect of the device. A retro-auricular implantation of other gravitational valves, such as GAV and PaediGAV (Aesculap, AG, Tuttlingen, Germany), is common in both adults and pediatric patients. However, a correct vertical alignment should always be ensured because the movement of the head with relation to the body axis might compromise the performance of the device. In adults, we prefer to implant GDs based on ball technology in the chest to avoid or minimize this problem (Figure 4A). Additionally, the M-DSV has been specifically designed to be implanted in the thorax. When this valve is implanted, the angle between the vertical axis of the valve should not deviate more than 15° from the gravitational axis to avoid interference with the mechanical design of the valve.

Another recommended precaution when implanting an M-DSV in the thorax is that it should be placed on a flat surface. In male patients, the valve can be placed at the infraclavicular or sub-mammary level. However, because of females’ variable breast size, it is better to place the valve at the sub-mammary level, on the costal grid, and at enough distance so that the bra does not coincide with the valve. Finally, given the dimensions and weight of the valve (28×6 mm, 11.6 g), it must be sutured in the muscular plane with a non-absorbable suture to ensure that the device does not move (Figure 4A).

## 9. Clinical Experience with M-DSV Use

The first clinical experience with this device was reported by Sprung et al. [7,19]. In 1997, who reported on implants conducted in 35 adults with hydrocephalus of different etiologies. In a randomized study, the first 21 patients were compared with 21 patients who received a conventional medium DPV, revealing that DPV patients had a significantly higher percentage of over-drainage-related complications, such as hygromas and hematomas [19]. Trost et al. (1998) [20] reported their experience of M-DSV implantation in 32 adult patients. The clinical results were excellent to good, except for three cases with previous severe post-traumatic or post-hemorrhagic brain damage. The authors remarked that despite achieving a satisfying clinical improvement in most cases, they saw only minimal to slight reduction in ventricular size. Additionally, apart from one case of non-symptomatic transient hygroma attributed to a suboptimal alignment of the valve with the gravitational axis, the authors observed no valve-related complications, such as over-drainage, underdrainage, or dysfunction. In two patients, however, three revisions were necessary due to disconnections of the peritoneal catheter [20]. Table 3 summarizes the abbreviations and complete names of the valves mentioned in this review.

Zeilinger et al. (2000) reported their clinical experience with the M-DSV in patients with NPH [21]. The authors treated 117 NPH patients, 47 with a Cordis-Standard DPV (CSV, Integra DP™ Valve System, INTEGRA LIFESCIENCES CORPORATION, Ratingen, Germany), 20 with a type 1 Orbis Sigma valve (OSV ITM Valve System, INTEGRA LIFESCIENCES CORPORATION, Ratingen, Germany), and 50 with an M-DSV. Ninety-five patients (39, 19, and 49, respectively) were reexamined after a mean time interval of seven months. No statistical differences in mechanical and infectious complications were found between the different models. However, the authors found significant differences in over-drainage complications and subdural hematomas, which occurred in 6% of the patients treated with the CSV, in 26% of those treated with an OSV, and 5% of those treated with an M-DSV. Three patients with an Orbis-Sigma valve developed subdural hematomas that required surgical evacuation. Two patients with an M-DSV also developed subdural hematomas; in one of these patients, the hematoma was resorbed without clinical symptoms, while in the other, the shunt had to be removed [21]. The authors concluded that the high incidence of over-drainage complications and subdural hematomas in the OSV group might contraindicate the use of this valve and that the hydrostatic M-DSV might be the best valve for NPH patients [21]. Similar conclusions were presented by Meier et al. [22] and later in a prospective multicenter study using the M-DSV in 128 patients with NPH [18].

A Japanese study demonstrated further interesting results derived from the use of M-DSV [23]. The authors implanted 94 M-DSVs in adult patients with hydrocephalus (10 cm H_2_O until April 2003 and 5 cm H_2_O from May 2003 in the supine position and 30 or 40 cm H_2_O in the upright position according to height) and carried out IVP monitoring via the shunt reservoir on all patients for one week after shunt placement. Regardless of the valve settings selected, IVP in the sitting position was slightly below zero in most patients, but no negative pressures below −15 cm H_2_O were observed in any of them [24]. Unfortunately, at least six patients who showed initial clinical improvement after shunting worsened due to underdrainage, despite a patent shunt. These patients had been implanted with a shunt set at an OP of 10 cm H_2_O for the supine position. In our experience, the best supine shunt OP for NPH is 5 cm H_2_O or even lower. On the other hand, chronic subdural hematoma occurred in three patients. The authors concluded that the M-DSV could maintain physiological IVP in patients with hydrocephalus, regardless of posture, and provide a generally satisfactory clinical outcome, but it could not prevent all over-drainage-related problems [24]. 

Finally, Hertel et al. reported on the widest clinical experience with the use of the M-DSV to date [15]. Between 1998 and 2005, these authors implanted an M-DSV in 169 adult patients (70 with NPH). They reported a rate of shunt response of 93.2%, an over-drainage rate of 3.2%, and no valve obstruction in the entire series, including patients with relatively high CSF protein content (mean protein content in CSF was 50.1mg (18–122 mg)). The overall shunt survival rates among these patients were 86%, 84%, and 82 % after 1, 2, and 6 years, respectively [15]. The overall complication rate for the study period was 21%, the most frequent complication being revision of the peritoneal catheter (8%), followed by infection (6%), over-drainage (3.5%), revision of the ventricular catheter (1%), and other complications (2.5%) [15]. In this study, it is important to note the high rate of problems with the distal catheter.

In our department, between October 2011 and November 2017, 127 ventriculoperitoneal M-DSV valves were implanted. Forty-two of them had an OP of 5/30 cm H_2_O, and in 85, the OP was 5/40 cm H_2_O.

## 10. The Problem of Catheter Dislocation/Fracture in the M-DSV

A specific problem associated with the use of VP or LP M-DSV is the potential fracture and disconnection of the proximal and/or distal catheters in the valve (Figure 4 and Figure 5). We have observed this problem in many units of this device after at least 10 months from implantation. Visual inspection of removed devices (Figure 5) showed that the catheters have an irregular fracture surface with well-defined edges and maintain the typical elasticity and rubbery feel of the original catheter. Frequently, when this problem occurs, it is necessary to replace the valve and the distal catheter. In our center, some of the explanted devices have been returned for a bench test of the valves’ performance (Figure 5). In all the explanted valves, the manufacturer reported that the bench test for all these valves was correct.

It has been suggested that catheter fractures in the M-DSV might be due to immunological reactions against the silicone of the subcutaneously-fixed catheter [25]. However, to my knowledge this phenomenon has not yet been described in other gravitational valves implanted in the skull, such as the GAV or PaediGAV shunt systems, despite that the same type of materials and connections as the M-DSV valves implanted in the chest are used. The most probable hypothesis to explain this phenomenon is the stress generated by the movement of two different materials with the high inertial loading of the M-DSV due to its unusual weight. The greater mobility of the valves placed in the thorax produces shearing forces in the small region of the catheter where it leaves the device.

The literature on catheter problems in hydrocephalus valves focuses mainly on the degradation, calcification, and aging of the catheter material, which becomes brittle over time [26]. Catheter failure or dislocation accounts for approximately 3% of complications in the M-DSV and typically occur more than five years after implantation [25]. Mechanical stress on the catheter combined with material degradation eventually leads to catheter failure, often occurring at places in which the catheter is under great dynamic stress. Some of the valves implanted at our center, however, were explanted before 12 months after implantation (Figure 4), well below the typical degradation time for this type of silicone catheter, and the typical signs of silicone-degradation were observed in none of them. This, therefore, argues that are other mechanical factors that play a role in catheter’s breakage. The size and weight of the M-DSV probably plays a role in the mechanical forces that act at the points were the proximal and distal catheter are attached.

In our patients, the characteristics of the broken catheter tips suggest a mechanical failure of the catheter following contact with the connector. This type of fracture surface is typical of breakage due to pure mechanical stress (Figure 5) and might be amplified by chemical or enzymatical degradation or calcification of the catheter. A degraded catheter, however, would typically leave a brittle, crumbly surface, and an overall change of consistency when compared to the newly implanted material, characteristics that were not observed in any of the broken catheters. If one assumes the catheter material has not been degraded at the time of explantation, the failures reported in our patients suggest mechanical stress in the catheter generated by the interaction between it and the metallic connector. This conclusion, also proposed by Trost et al. [20], is supported by the fact that the medical device manufacturer released a second and third generation of the device (Figure 6), improving, among other aspects, the design and mechanical robustness of the catheter-connector linkage. Although the manufacturer reinforced the material of this critical region, the first and second generation of these valves remain in use and must be controlled periodically. In lumbo-peritoneal valves, the manufacturer has not yet implemented this improvement. Furthermore, no long-term studies on third-generation valves have been carried out.

## 11. Final Considerations and Recommendations

According to Aschoff et al. [27], an ideal shunt system should be: (1) effective in all body positions with compensation of postural changes in hydrostatic pressures; (2) resistant to movements and vibrations of daily life; (3) insusceptible to external pressure, flexion, and torsion; (4) able to effect sufficient drainage of flow peaks (e.g., nocturnal); (5) insensitive to increased content of protein and cellular material; and (6) able to retain the desired properties for a very long time. Added to these characteristics would be the possibility of adjusting the opening pressure of the valve. Although this ideal shunt system has not yet been developed, the M-DSV has many of these qualities.

The hydrodynamic properties of the M-DSV promise to prevent posture-related over-drainage. In comparison to ASD, gravitational valves have the important advantage of being unaffected by external or subcutaneous pressure [9]. Research (both by the manufacturer and independently) carried out with small (10 microns diameter) and big particles (diameter greater than 25 µm) has shown that the M-DSV has a better tolerance to proteins and small quantities of blood in the CSF when compared with other valves [9,10]. This makes the M-DSV ideal for implantation in patients with high levels of proteins and cells in the CSF. However, if we have implanted M-DSV, especially of the first and second generation without reinforcement at the junction point between the valve and the distal catheter (Figure 6C), the follow-up of patients should include X-rays for detecting potential catheter fracture at this point over time.

To conclude, based on the information reported in the present paper, a series of recommendations must be considered when selecting the M-DSV:The M-DSV is s hunt device that controls posture-related over-drainage.The M-DSV is designed for adults and should not be implanted in children, as they always have greater body mobility than adults.When implanting the M-DSV valve, the flattest part of the chest should be selected, and the device correctly fixed to the muscular plane so that the vertical direction of the valve is in line with the vertical axis of the patient’s body.In terms of sensitivity to changes in CSF viscosity, composition, and sticking, as well as the influence of the pressure of the subcutaneous tissues, the M-DSV has superior performance compared with other valves [6,7].Due to the risk of fracture in the catheters at the level of the proximal or distal connectors of the valve, it is advisable to include in the patient’s follow-up periodic radiological controls for ruling out this potential problem.

## Figures and Tables

**Figure 1 jcm-10-00246-f001:**
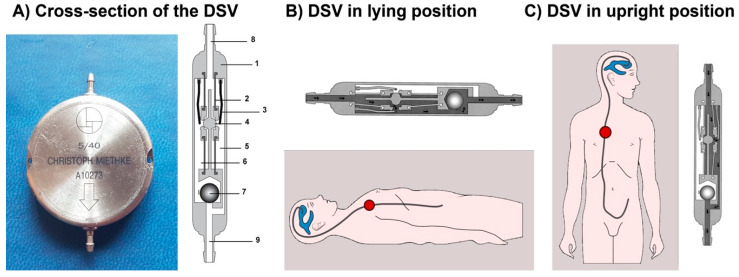
(**A**) Cross-section of the Miethke^®^ Dual-Switch valve (DSV). It has a solid titanium casing (1). Two titanium plates (3) are integrated into diaphragms made of silicone (2). Each plate, together with a ball (4), creates a valve seat, which is integrated into the casing as an opening and closing mechanism. Two different springs control the position of the plates. There is a stronger spring for the high-pressure chamber (5) and a weaker one for the low-pressure chamber (6). A heavy tantalum ball (7) is used to shift the flow path to either chamber. Titanium lips are used for the inlet (8) and outlet (9) tubing connection. When patients are supine (**B**), the low-pressure chamber operates like a conventional DPV and is activated when the differential pressure is above the opening pressure of the valve. In the upright position (**C**), the low-pressure valve is closed, and the postural changes move a tantalum ball that opens the flow path for the high-pressure controller [7].

**Figure 2 jcm-10-00246-f002:**
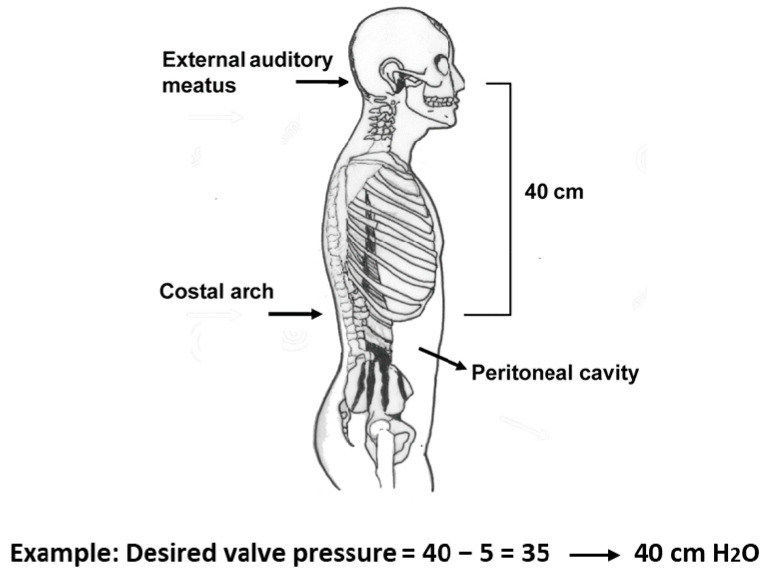
Method of selecting a suitable pressure for the high-pressure valve in the Miethke^®^ Dual-Switch valve. (1) Measure the distance between the third ventricle (external auditory meatus) and the patient’s diaphragm (costal arch); (2) subtract 5 cm from the measured distance; (3) choose a valve whose high-pressure setting exceeds the final measured value by the smallest amount.

**Figure 3 jcm-10-00246-f003:**
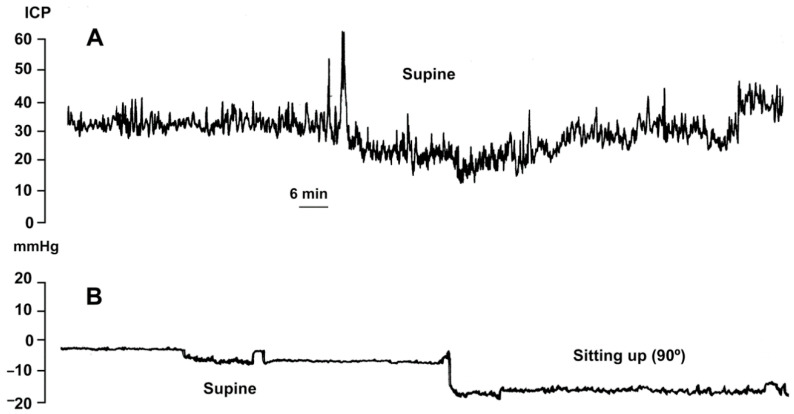
Intracranial pressure (ICP) readings from a patient with idiopathic intracranial hypertension before (**A**) and after the implantation of a 10/40-cm H2O Miethke^®^ Dual-Switch valve (**B**). In the posture-induced ICP changes after shunting (**B**), the patient remained supine for 1 h and then sat up and remained sitting for 3 h.

**Figure 4 jcm-10-00246-f004:**
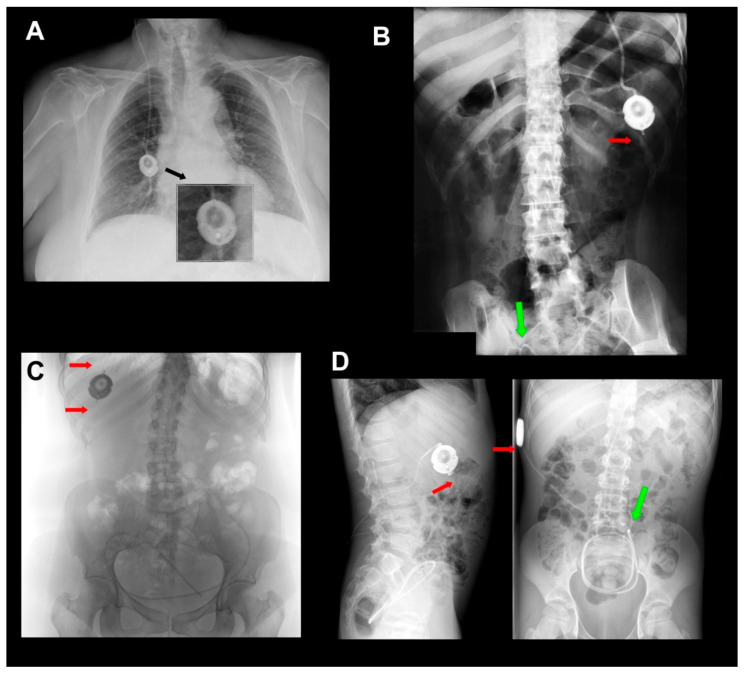
(**A**) A 10/50-cm H2O Miethke^®^ Dual-Switch valve (M-DSV) implanted in a 75-year-old woman affected by normal pressure hydrocephalus (NPH). Note how the vertical axis of the valve is perfectly aligned with the vertical axis of the patient’s spine. This is the recommended position when this type of valve is inserted. The device is anchored to the muscular plane by a 0-silk suture. (**B**,**C**) Chest X-ray of the thoracic implantation site of the M-DSV in several patients showing fractures of the catheters (red arrows). (**B**) Fracture of the distal catheter of the valve in a patient aged 23 years with hydrocephalus associated with spina bifida with abdominal migration of the catheter (green arrow). A fracture of the distal catheter was diagnosed six years after implantation of the valve. (**C**) Fracture of the proximal and distal catheters of the valve in a patient with hydrocephalus associated with a Chiari type 1 malformation (red arrows). (**D**) Early distal catheter fracture (red arrow) when using a lumbo-peritoneal M-DSV. Although it would be expected that this problem would happen a few years after the implantation of the valve, in our center, we observed this problem only ten months after valve implantation in a 12-year-old child. The valve was replaced, and the child presented this problem for the second time one year after this second implantation. Greater mobility in children, when compared to adults, reinforces the fact that this type of valve has been designed for adults and should not be implanted in children.

**Figure 5 jcm-10-00246-f005:**
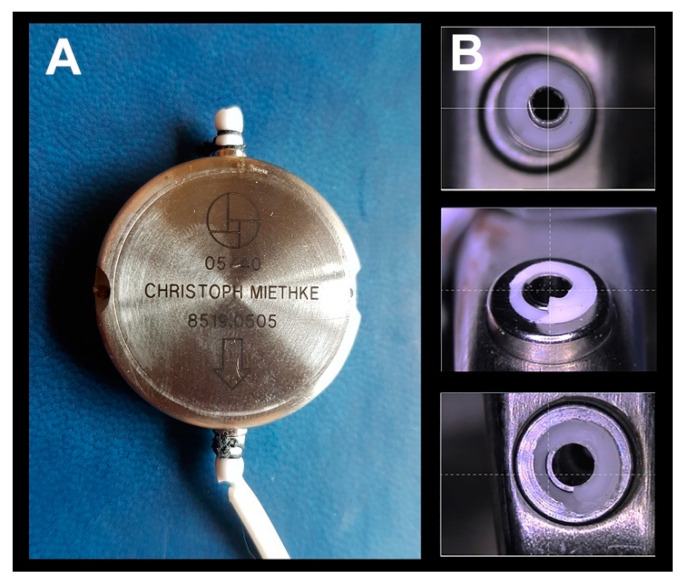
Images of Miethke^®^ Dual-Switch valves (M-DSV) explanted after diagnosing a fracture of the distal catheter of the valve. Fractures typically occur at the connector tip level of first (**A**) or second (**B**) generation M-DSV. Note that at the distal end of the valve, there is a catheter fragment sutured to the connector, indicating that a fracture of the catheter at the distal end of the connector had occurred and not a disconnection of the catheter due to failure of the suture.

**Figure 6 jcm-10-00246-f006:**
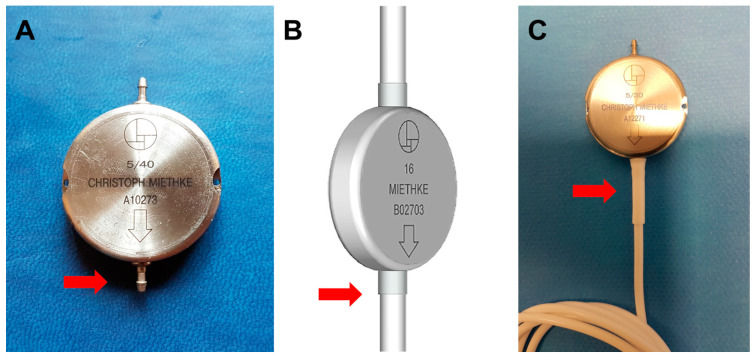
Schematic diagrams and real images of the first (**A**), second (**B**), and third (**C**) generation of the Miethke^®^ Dual-Switch Valve. Among other aspects, the most important design improvement was the mechanical robustness of the catheter-connector linkage (arrows).

**Table 1 jcm-10-00246-t001:** Closing pressures stated by the manufacturers (left) compared to the pressure values found in the tested shunts by Czosnyka at the UK Shunt Evaluation Laboratory (right, in parenthesis) for different models of Dual-Switch valves.

Pressure Settings for Ventriculoperitoneal Shunts			
Low Pressure (Supine Position) mm H_2_O	High Pressure (Upright Position) mm H_2_O	Low Pressure (Supine Position) mm Hg	High Pressure (Upright Position) mm Hg
5 *	30	3.7	22.0
5 *	40	3.7	29.4
5 *	50	3.7	36.8
10	30	7.4/(7 ± 0.6) **	22.0/(22 ± 2.1) **
13	30	9.5/(9.5 ± 0.4) **	22.0
16	30	11.8/(12 ± 1.1) **	22.0
10	40	7.4	29.4/(28 ± 1.9) **
13	40	9.5	29.4
16	40	11.8	29.4
10	50	7.4	36.8/(38 ± 2.2) **
13	50	9.5	36.8
16	50	11.8	36.8
**Pressure Settings for Lumboperitoneal Shunts**			
10	30	7.4	22.0
10	40	7.4	29.4
10	50	7.4	36.8

* Valves especially designed for patients with normal pressure hydrocephalus. Adapted from the UK Shunt Evaluation Laboratory Report [9]. ** Pressure values found in the tested shunts by Czosnyka et al. [10] Values here are given as mean ± standard deviation.

**Table 2 jcm-10-00246-t002:** Evaluation of the performance and hydrodynamic properties of Miethke^®^ Dual-Switch valves (M-DSV) by the U.K. Shunt Evaluation Laboratory.

**Mechanical Durability**	All the shunts showed mechanical durability over the period of testing (>48 days) and good stability of hydrodynamic performance over a 28-day period.
**Opening and Closing Pressures**	Very well defined and stable in time, linearly changing with performance level. The distal catheter did not change the opening and closing pressure significantly.
**Pressure-Flow Performance Curves (PFPC)**	The PFPC, operating, opening, and closing pressures were stable, with minimal scatter, and were in accordance with data reported by the manufacturer for valves working in both horizontal and vertical positions.
**Hysteresis of the PFPC**	There was no hysteresis of PFPC. Slight hysteresis (but very narrow, 1 mm Hg of width) appeared after connection of the distal catheter.
**Pressure Changes Related to Body Posture**	The valves showed increased stepwise operational pressure in vertical position when compared to horizontal position, according to its fixed parameters.
**Hydrodynamic Resistance**	The valves have rather low and stable hydrodynamic resistance (2.2 ± 0.8 mm Hg/mL/min) and therefore are able to stabilize ICP according to the fixed settings. The addition of a distal catheter increases the shunt resistance to 2.9 ± 1.1 mm Hg/mL/min.
**Effect of Particles in Reagent**	Small particles (10 microns diameter) temporarily increased the valve’s hydrodynamic resistance to 7 mm Hg/mL/min. Large microspheres (with a diameter greater than erythrocytes—25 µm) were found to open the shunt temporarily (closing pressure decreased from around 9 to 3 mm Hg). In both cases, these effects were only temporary when the particles were washed out spontaneously and on the next day the valve returned to normal performance.
**ISO/DIS 7197 Compliances**	The M-DSV complied with this international standard for the testing of hydrocephalus valves.
**Resistance to Breakage and Leakage of Preassembled Junctions**	Preassembled junctions did not break when a test specimen was subjected to a load of 1 Kg of force for 1 min. All junctions remained free from leakage when the water pressure was increased to 3 kPa (about 25 mm Hg).
**Reflux**	The valves did not show any reflux when tested according to the ISO standard. Valves did not exhibit flow reversal for an outlet-inlet differential pressure of up to 100 mm Hg.
**Magnetic Influence**	Magnetic field 3T did not have any influence on the shunt’s performance.
**External Pressure Influence**	No change in operating pressure occurred when an environmental pressure of 7 mm Hg was applied.
**Temperature Influence**	None of the parameters (opening, closing pressure, and resistance) were altered by a temperature change of 30 °C to 40 °C.

Adapted from the UK Shunt Evaluation Laboratory Report [9,10].

**Table 3 jcm-10-00246-t003:** Abbreviations and complete names of the valves mentioned in this review.

Abbreviation	Complete Name	Distributed by:
**CSV**	Cordis-Standard differential-pressure valve	INTEGRA LIFESCIENCES CORPORATION, Ratingen, Germany
**DPV**	Differential-pressure valve	
**GAV**	MIETHKE^®^ gravitational valve	Aesculap, AG, Tuttlingen, Germany
**PaediGAV ***	Pediatric MIETHKE^®^ gravitational valve	Aesculap, AG, Tuttlingen, Germany
**M-DSV**	Miethke^®^ Dual-Switch valve	Aesculap, AG, Tuttlingen, Germany
**OSV**	Orbis Sigma valve	INTEGRA LIFESCIENCES CORPORATION, Ratingen, Germany

* The MIETHKE paediGAV^®^ is the smallest gravitational valve for the treatment of pediatric hydrocephalus.

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
