# Peer review of "Considerations in the Use of Gravitational Valves in the Management of Hydrocephalus. Some Lessons Learned with the Dual-Switch Valve"

_jcm, 2021, doi:10.3390/jcm10020246_

Round 1
Reviewer 1 Report
This is a review/personal experience/retrospective single center analysis written by both guest editors of the special issue "Diagnosis and Treatment of Normal-Pressure Hydrocephalus Syndrome—an Update".
While I do not have criticisms on the paper I find it surprising to review a paper that has already a citation remark for the next year in this journal.
Author Response
Reviewer #1: This is a review/personal experience/retrospective single center analysis written by both guest editors of the special issue "Diagnosis and Treatment of Normal-Pressure Hydrocephalus Syndrome—an Update".
While I do not have criticisms on the paper I find it surprising to review a paper that has already a citation remark for the next year in this journal.
Response:We do not understandReviewer #1’s comment regarding the ‘citation remark’. However, he/she did not request any specific changes. We appreciate that he/she took the time to review the article.

Reviewer 2 Report
In the present study, the authors have addressed management of hydrocephalus in patients with NPH using gravitational valves. In particular, the main features of the Miethke® dual-switch valve, including its main advantages and disadvantages, were discussed with the aim of providing recommendations for its use.
The authors have described in detail the features and clinical use of The Miethke® Dual-Switch Valve. In addition, the neurosurgical intricacies of implanting this type of valve have been clearly described so that this can be used as a technical note.
Although this valve type is no longer used in our clinic (pro-SA current), we also have few complications to report with this valve.
Despite the advancement of technical improvement of valves for patients with hydrocephalus, The Miethke® Dual-Switch Valve may well be used in some cases with a good clinical outcome.
Summary:
Good review work-> recommendation for publication.
Author Response
Reviewer #2: In the present study, the authors have addressed management of hydrocephalus in patients with NPH using gravitational valves. In particular, the main features of the Miethke® dual-switch valve, including its main advantages and disadvantages, were discussed with the aim of providing recommendations for its use.
The authors have described in detail the features and clinical use of The Miethke® Dual-Switch Valve. In addition, the neurosurgical intricacies of implanting this type of valve have been clearly described so that this can be used as a technical note.
Although this valve type is no longer used in our clinic (pro-SA current), we also have few complications to report with this valve.
Despite the advancement of technical improvement of valves for patients with hydrocephalus, The Miethke® Dual-Switch Valve may well be used in some cases with a good clinical outcome.
Summary:
Good review work-> recommendation for publication
Response:We thank Reviewer #2 for his/her nice comments. We appreciate his/her review of the article. Our learning curve with gravitational devices was with the Dual-Switch valve, and we believe some of the lessons learned by us could be useful for neurosurgeons managing hydrocephalus with any gravitational valve.

Reviewer 3 Report
This is a very interesting and comprehensive paper that should be published after revisions.
Please see uploaded mark-up draft of the paper with suggestions and questions.
Early on, while this may be obvious to them, the authors should emphasize to the reader what type of valve is usually used, and why their system is dual. They should also outline what they will cover in their paper.
In several places, if there are more updated references, the authors should cite them.
Figure 1 is nice, but it might help the reader to also have a schematic of a patient lying and standing with a standard valve operating, vs. their dual valve.
The message from Table 1 needs to be clarified. Is this to compare the manufacturer's specifications versus the readings actually obtained?
Table 2 is extensive. Has this already been published?
The paper seems to start out as a review, but then morphs into a clinical study, especially concerning the issue of catheter fracture. Over what period of time were these cases collected? It should be made clear that the information is exempt from IRB consideration, if this is indeed true.
It is not always clear to the reader the distinctions between DPV, OSV and M-DPV for instance on page 10. Possibly a table might help with understanding this data.
I cannot see the fractures in Fig. 4.
Section 10 could be more concise.
I think the overall advice / conclusions should be expanded to include their experience and recommendations about why neurosurgeons should use this particular valve (or in which patients) and highlight the most important considerations (Section 11, paragraph 2). What in fact are the considerations and lessons learned, briefly?

Author Response
Reviewer #3:
Point 1:This is a very interesting and comprehensive paper that should be published after revisions.
Response 1.We are grateful to Reviewer #3 for the opportunity he/she offers us to improve our manuscript by addressing his/her thoughtful comments.
Point 2.Please see uploaded mark-up draft of the paper with suggestions and questions.
Response 2.We appreciate the reviewer's suggestions and our changes are included in the revised version of the manuscript.
Point 3.Early on, while this may be obvious to them, the authors should emphasize to the reader what type of valve is usually used, and why their system is dual. They should also outline what they will cover in their paper.
Response 3. In accordance with the reviewer's suggestions, the content of the introduction has been modified (changes are marked in the revised version of the manuscript).
Point 4:In several places, if there are more updated references, the authors should cite them.
Response 4.Although some references may seem old, they correspond to classic articles that we believe are crucial for following the development of gravitational devices in the management of hydrocephalus. We sought to include the most relevant articles related to the Dual-Switch valve in the article, but if Reviewer #3 believes we have missed any important ones, we will be happy to incorporate them.
Point 5.Figure 1 is nice, but it might help the reader to also have a schematic of a patient lying and standing with a standard valve operating, vs. their dual valve.
Response 5. We thank the reviewer for his/her comments. We have changed the Figure 1 and we hope this clarifies how the valve works when the patient is recumbent and, in a sitting, or standing position.
Point 6.The message from Table 1 needs to be clarified. Is this to compare the manufacturer's specifications versus the readings actually obtained?
Response 6. The reviewer is correct. To clarify this, the title of Table 1 has been changed to “Closing pressures stated by the manufacturers (left) compared to the pressure values found in the tested shunts by Czosnyka at the UK Shunt Evaluation Laboratory (right, in parenthesis) for different models of Dual-Switch valves”
Point 7. Table 2 is extensive. Has this already been published?
Response 7. Table 2 is not a copy of what has been previously published by Czosnyka’s group. In brief, it reflects a synthesis of the most relevant aspects related to the Dual-Switch valves found in different articles published by the Cambridge Laboratory by these authors. Because the article is intended to be a thorough review of this type of valve, we believe that it is important to preserve this table, or alternatively move it to a supplement, since it will allow readers to find all the most relevant technical information about the Dual-Switch valve.
Point 8. The paper seems to start out as a review, but then morphs into a clinical study, especially concerning the issue of catheter fracture. Over what period of time were these cases collected? It should be made clear that the information is exempt from IRB consideration, if this is indeed true.
Response 8. The article is a review of the characteristics of the Dual-Switch valves, in which we introduced notes related to our personal experience, without actually being a clinical study. The information we provide arises from a series of patients treated between October 2011 and November 2017: 127 ventriculoperitoneal M-DSV valves. This information is included in the last sentences of section 9: “In our department, between October 2011 and November 2017, 127 ventriculoperitoneal M-DSV valves were implanted. Forty-two of them had an OP of 5/30 cmH2O, and in 85, the OP was 5/40 cmH2O” (lines 376-378). This information and some of the images included in the review (Figures 3-5) are the only personal information included in the paper. After consulting the ethics committee of our center, they concluded that the paper could be exempt from IRB approval.
Point 9. It is not always clear to the reader the distinctions between DPV, OSV and M-DPV for instance on page 10. Possibly a table might help with understanding this data.
Response 9. Reviewer #3 is right. Per his/her suggestions, we have incorporated a table in which we define all the abbreviations of the valves mentioned in the review (Table 3: Abbreviations and complete names of the valves mentioned in this review).
Point 10. I cannot see the fractures in Fig. 4.
Response 10. Figure 4 shows fractures in the valve catheters shown in images B, C, and D with some dislodged abdominal catheters. To make it more obvious, we have added red arrows at each fracture point and green arrows in the dislodged abdominal catheters.
Point 11. Section 10 could be more concise.
Response 11. From our perspective, one of the relevant parts of this review is the problem of fracture of the proximal catheter, but more frequently of the distal catheters, which occurs over time in the first and second generation of these valves. We believe that the neurosurgeons who have implanted them should be aware of this and closely follow-up with these patients. In our center, we periodically see this type of problem with patients treated a few years ago. Because of this, we would appreciate if Reviewer #3 would allow us to leave the text as it is now in the revised version of the manuscript with minor deletions.
Point 12. I think the overall advice / conclusions should be expanded to include their experience and recommendations about why neurosurgeons should use this particular valve (or in which patients) and highlight the most important considerations (Section 11, paragraph 2). What in fact are the considerations and lessons learned, briefly?
Response 12. Our main point is that neurosurgeons should use valves with gravitational control but not the Dual-Switch valve. In accordance with Reviewer’s suggestion, however, we have modified the last section of the article. We hope this clarifies this point.
